# Bovine schistosomiasis in some selected areas of South Wollo and Oromia Zones of Amhara Region, North-East Ethiopia

Gashaw Molla[1], Tarekegn Tintagu[2]*, Ahmed Yasin[2], Bethelehem Alemu[2], Alula Alemayehu Assen[2], Kassahun Tadesse[2]

1 Dessie Zuria Livestock Development Office, Dessie, Ethiopia, 2 School of Veterinary Medicine, Wollo University, Dessie, Ethiopia

* drtarekegn@yahoo.com

**Data Availability Statement:** All the data used for this research have been provided as Supporting information.

## Abstract

A cross-sectional coprological and pathological study was conducted in five districts of South Wollo and Oromia Administrative Zones, Amhara regional state, Ethiopia from November 2020 to June 2021 to determine the prevalence and associated risk factors of Bovine Schistosomiasis and to characterize pathological lesions induced by the adult worm of *Schistosoma bovis*. For coprological examination, a total of 768 fecal samples were collected both from the field (384) and the abattoirs (384). An abattoir survey was carried out on 384 cattle to evaluate the performance of the sedimentation method. The risk factors were identified using multivariable mixed-effect logistic regression analyses. The diagnostic efficacy of the sedimentation technique was determined by calculating sensitivity and specificity considering postmortem examination as a reference test. The overall prevalence of bovine schistosomiasis using coprological examination was found to be 16.7% (95% CI = 14.10–19.49). The prevalence of shistosomiasis based on post-mortem examination was found to be 17.19% (95% CI = 13.55–21.34). Local cattle breed (OR = 2.44, 95%CI = 1.34–4.43), poor body condition (OR = 4.09, 95% CI = 2.45–6.83) and adult (OR = 1.78, 95% CI = 1.21–3.28) cattle are more likely to acquire shistosomiasis than crossbreed, good body condition, and young cattle. The sensitivity and specificity of sedimentation techniques, keeping postmortem examination as a reference test were 74.24% (95%CI = 61.99–84.22) and 98.11% (95%CI = 95.94–99.30), respectively. The major gross lesions were observed in the liver and intestinal tracts. In conclusion, adult local cattle with medium and poor body conditions should be prioritized for deworming and future surveillance.

## Introduction

Ethiopia has the largest number of livestock in Africa with an estimated of 70 million cattle, about 42.9 million sheep, 52.5 million goats, 2.15 million horses, 10.80 million donkeys, 0.38 million mules, about 8.1 million camels and 48.9 million Poultry [1]. Livestock is a major source of animal protein, power for crop cultivation, means of transportation, export

**Funding:** There was no finacial funder to this research work.

**Competing interests:** All the authors declared that there is no competing interest.

commodities, manure for farm land and household energy, security in times of crop failure, and means of wealth accumulation. The sector contributed up to 40% of agricultural Gross Domestic Product (GDP), nearly 20% of total GDP, and 20% of national foreign exchange earnings in 2017. The Ethiopian livestock population is almost entirely composed of indigenous animals. Recent estimates showed that 97.8%, 1.9%, and 0.3% of cattle are indigenous, hybrid, and exotic breeds, respectively [2].

However, the livestock sector is not efficiently and fully exploited due to several constraints like malnutrition, traditional husbandry practice, poor genetic makeup and prevailing diseases [3, 4]. In addition, the existing veterinary service delivery under the current animal health system is considered to be unsatisfactory in both private and public sectors because of the low government attention, inadequate budget allocation, gaps in control of illegal drug circulation, implementation of ethical practices, and shortage of basic laboratory facilities [5]. As a result, 60 million poultry and 5–6 million cattle are lost every year due to poor animal health service and the young stock mortality which has a huge economic impact and becomes a trade barrier [2].

Parasitic diseases have been indicated among the major constraints in animal production throughout the tropical and sub-tropical countries of the world. Trematodes in general and schistosoma in particular have been reported as one of the major problems both in animals and humans around the world [6, 7]. The geographical distribution of bovine schistosomiasis has been determined primarily by the distribution of snail intermediate host particularly *Bulinus* species which are important for the occurrence of disease in bovine species. It has been indicated as endemic in sub-Saharan Africa and also found commonly in northern, eastern, southwestern and central parts of Ethiopia [8–10].

The endemicity of the disease particularly in the area with large permanent water bodies and marsh pasture area which anticipated for being ideal for the propagation and maintenance of the intermediate host (snails) has been found responsible for high prevalence of the infestation [11, 12]. Out of 10 species reported as naturally infesting cattle, six have received particular attention mainly because of their recognized veterinary significance: *Schistosoma bovis*, *S. indicum*, *S. japonicum*, *S. matthei*, *S. intercalatum*, *S. nasale* and *S. rodhoni* [7].

The course of infestation is often divided into three phases: migratory, acute and chronic. The migratory phase occurs when cercariae penetrate and migrate through the skin. This is often asymptomatic, but in sensitized patients, it may cause transient dermatitis and occasionally pulmonary lesions and pneumonitis [13]. Eggs released into the bloodstream by adult worms can invade local tissues, where they release toxins and enzymes and provoke a TH-2-mediated immune response [14]. Inflammation and granuloma formation occurs around deposited eggs, which can lead to fibrosis and scarring of affected tissues, if the burden of eggs is heavy. In the bowel, granulomatous inflammation around the invading eggs can result in intestinal schistosomiasis characterized by ulceration and scarring [15]. At necropsy during the acute phase of the disease there is marked hemorrhagic lesions in the mucosa of the intestine, but as the disease progress the whole of intestine appears grayish, thickened and edematous due to confluence of the egg granuloma and the associated inflammatory changes and in the chronic stages of the disease the pathology is associated with collagen deposition and fibrosis, resulting in organ damage and dysfunction [16].

Current diagnosis of *S. bovis* infestation includes evaluation of clinical signs, pathological lesions, parasitological and serological techniques. As clinical signs caused by *S. bovis* are similar to those produced by other trematode parasites, confirmation of *S. bovis* infection under field condition by these method is unreliable [17]. At necropsy, S. bovis infestation can be diagnosed by finding thousands of visible adult worms in the mesenteric vessels. Infected livers are diagnosed on the basis of the presence of macroscopic lesions of schistosomiasis visible as

white-gray foci under the liver capsule and within the substance of the liver [18]. Definitive diagnosis of an active *S. bovis* infection can be made only by detecting eggs of the parasite in faeces or biopsy specimen of the infected animal [19].

The disease has been indicated to be responsible for considerable economic losses in live-stock industry, mainly through mortality, reduced fertility, productivity and stunted growth [20, 21]. Schistosomiasis control focuses on reducing disease through periodic, large-scale population treatment with praziquantel; a more comprehensive approach including potable water, adequate sanitation, and snail control [22]. Ethiopia is highly endemic for Schistosomiasis, since temperature in Ethiopia appears to be the major factor that affects the distribution of Schistosoma species [23].

The prevalence of *Schistosoma bovis* has been reported from different regions of Ethiopia. Accordingly, prevalence of 28% in kemisse [24], 10.17% in Fogera [25], 13.70% in Fogera [12], 7.6% in Debretabor [26], 11.5% in Dangila [20], 26.3% in Bahir Dar [27], 48% in Bahir Dar [28], 30.3% in Bahir Dar [10], 28.14% in Bahir Dar [29] and 13.46% in Jimma [11] have been reported both by coprological and postmortem examinations.

Even if different research findings reported from different parts of the country, considering persistent water bodies of the current study areas, research has not been conducted previously. Therefore, considering the economic importance of the disease, agro ecological variation, different animal husbandry practices and disease prevention and control strategy of the five districts, the current study was designed to determine the prevalence of *S. bovis* with the associated potential risk factors in the five study districts of south wollo and oromia special zones and to characterize pathological lesions induced by *S. bovis* infestation as well as to evaluate the efficacy of sedimentation technique with respect to postmortem examination.

## Materials and methods

### Study area description

The study was conducted in the selected districts of South Wollo Zone such as Kalu, Tehuledere and Ambasel Districts around Ardibo, Logo and Golbo lakes and Borkena river) and Oromia special Zone of Dawa Cheffa and Artuma furci. Those study areas were selected purposively based the availabilities of permanent water bodies. Kalu District is one the administrative districts in south Wollo Zone, Amhara Regional State. The district consists of 34 kebeles (30 rural and 4 urban Kebeles). The elevation ranges from 1400 to 1850 meters above sea level. There is bimodal raining pattern in the district and the mean annual rainfall ranges from 700 to 900 mm. The relative humidity of the area varies from 23.9% to 79% and the average minimum and maximum daily temperatures are 11.7˚C and 23.9˚C, respectively [30].

Tehuledere District is among the districts of South Wollo Zone which is located in north eastern Ethiopia. The district is found 431 Km away from the capital of Ethiopia, Addis Ababa and 540 km east of Bahir Dar Town, regional capital city. The district located at 11˚17'N latitude and 39˚ 42'E longitude, and the elevation of the district ranges from 500 to 2700 meters above sea level with 1200 mm annual rainfall [31].

Ambasel District is one of the 24 administrative districts in south Wollo Zone, Amhara Regional State which is located at a distance of 420 km from Bahir Dar, the regional capital city and 460 Km North of Addis Ababa, capital city of Ethiopia. The district is located at 11˚31'05" North latitude and 39˚36'34" East longitude with an elevation ranges from 1200 to 3200 meters above sea level. The average temperature of the district is 19˚c and mean annual rain fall (RF) is 500–800 mm which rains in bimodal pattern [32].

Oromia Special zone is found in Amhara Regional State, north eastern Ethiopia. It is located at 325 km North of Addis Ababa, the capital city of Ethiopia. Currently, seven districts are

found in this special zone including Dawa Chefa, Bati, Jile Timuga, Artuma Fursi, Dawa Harewa, Bati and Kemise towns. The geographical location of this zone is between 10˚01' N to 11˚25'N and 39˚ 41'E to 40˚24'E with elevation ranges from 1000 to 2500 meters above the sea level. The mean annual rainfall of the area ranges from 600 mm to 900 mm, and the minimum and maximum temperatures are 12˚C and 33˚C, respectively [33].

## Study design and population

Cross-sectional study was conducted from November 2020 to June 2021. Without discrimination of their sex, age, origin, breed and body condition cattle originated from the five purposely selected study districts were the study animals. The age, sex, body condition, breed and origin of the study animals were recorded as explanatory variables. The age of the individual animal was determined based on their dentition and categorized into three groups: young (<1year), adult (2–5 years) and old (> 5 years) [34]. Breeds were distinguished by their uniquely observable characteristics as described by Bosso et al. [35]. Body condition score were recorded as poor, medium and good [36].

## Sample size determination and sampling techniques

To calculate the total sample size, the following parameters were used: 95% confidence interval, 5% desired level of precision and 50% expected prevalence of bovine schistosomiasis among cattle in the study area since there was no previous work in the study sites, and the formula given by Thrusfield [37] was used.

$$n = \frac{1.96^2 * P_{exp}(1 - P_{exp})}{d^2} \tag{1}$$

Where n = required sample size

   Pexp = expected prevalence (50%)

   d = desired absolute precision (0.05)

   Accordingly, n = 1.962 *0.5 (1–0.50)/0.0025 = 384 cattle but, to increase the precision 768 cattle were sampled both from the field and the abattoirs.

   In this study mixed sampling techniques were employed to select the sampled animals. Sampled animals in the field were selected using simple random sampling technique whereas cattle presented for slaughtering from the specified study areas were selected purposively and listed and then systematic random sampling technique with an interval of five was used.

## Faecal sample collection and transportation

Both in the field and abattoirs the study animals were restrained properly and approximately 10 grams of fresh faecal sample was collected directly from the rectum using gloved hands and then the collected faecal samples were transported with cold chain to Wollo University, School of Veterinary Medicine, Parasitology laboratory.

## Parasite egg detection

For detection of the fecal egg excretion sedimentation technique was used. The fecal samples were concentrated using standard sedimentation technique described by Gupta and Singla [38]. 3 g of faeces placed into the flask and mixed with 42 ml of water 14 times their volume of normal saline and the mixture allowed to settle in a glass urine flask. The supernatant is removed after half an hour and the sediment suspended. This step is repeated until the

supernatant becomes clear. A drop of the final sediment is placed in a slide with a cover slip and examined under the microscope. Finally *S. bovis* egg has characterized photographically.

## Abattoir survey

Dessie and Kombolcha municipal abattoirs are small sized abattoirs located in South Wollo administrative zone in Dessie and Kombolcha towns, respectively. Their daily average slaughtering capacities are 30 and 25 heads respectively. Abattoir survey on *S.bovis* infestation in 384 cattle was carried out at both municipal abattoirs. Cattle came from various livestock markets from the selected districts for slaughtering were subjected to coprological examination for the presence and absence of schistosoma eggs and then during post-mortem examination all cattle were diagnosed for the presence and absence of adult schistosomes in their mesenteric veins and morphologically characterized using stereomicroscope [39].

Grossly lesions were identified by visualization, palpation and incision during necropsy [40]. The gross lesions were noted in liver and intestinal tracts and changes in affected organs were conducted depending on their size, color, consistency and shape, as well as other gross abnormalities and representative pieces of tissue from both organs were taken for histopathological examination.

## Ethical considerations

An official letter of cooperation was obtained from Wollo University ethical clearance committee dated on 12, October, 2020 with a reference number of WU/CMHS13/105/2020. The purpose of the investigation was explained to all eligible animal owners and informed consent was sought from all eligible animal owners to allow their animals for investigation before any procedure is performed. During specimen collection both in the field and the abattoirs all applicable international, national, and/or institutional guidelines for the care and use of animals were followed.

## Data management and analysis

The collected data was stored in the Microsoft excel database system for data management.

Statistical analyses were conducted using R version 3.6.1 [41]. The outcome for the model was the individual animal status for bovine shistosoma (positive or negative for shistosoma). In addition, the sex, age, breed and body condition of the study animals were included as potential risk factors (fixed effects) in the model. Furthermore, the interactions between these variables were included in the model. The origin of the animal where the samples were collected was included as a random effect to account for clustering, and the intraclass correlation (ICC) coefficient for the random term was calculated based on the methodology described for ICC estimation from the random intercept logistic model [42]. Clustering was deemed high for a random effect that had an ICC greater the 0.3 [43].

The study employed the mixed-effects logistic regression model approach under the generalized linear mixed models (GLMMs) fit by maximum likelihood which accommodates both random and fixed effects for the dichotomous dependent variable. Univariable mixed-effect logistic regression analysis was fitted to the data to quantify the effect of each explanatory variable on the dependent variable. Any variables with a *p*-value of less than 0.25 and not collinear with each other were then used to fit a multivariable mixed-effects logistic regression model shown in Eq 2 using package *lme4* [44].

A backward stepwise elimination approach was used until all variables had a p-value of <0.05 and were considered significantly associated with the outcome variable. Confounding was considered present if there is a 25% change in the coefficients of any of the remaining

variables after removing a non-significant value ($p > 0.05$) from the model. Goodness-of-fit of the final regression model was assessed by calculating conditional $R^2$ for the final model ($R^2_{GLMM(c)}$) and the amount of variation in the data explained by the fixed effects were determined by calculating marginal $R^2$ for the fixed effects ($R^2_{GLMM(m)}$) using package *performance* [45]. The final model included the effects under analysis and all other significant main effects without interaction effects since they are not significant. The final model was chosen based on the lowest Akaike Information Criterion (AIC). The collinearity analysis to detect the overlapping predictor variables on the outcomes, and the variance inflation factor (VIF) >10 or tolerance <0.1 was a criteria of severe collinearity [46]. The coefficient estimates and odds ratios and 95% CI were obtained for all classes within variables included in the final model. Estimated prevalence and confidence intervals were calculated using the *prevalence* package (v0.4.0) [47]. A p-value < 0.05 was considered statistically significant. The model fitted to the data is presented below:

$$logit(\boldsymbol{p_{ij}}) = \boldsymbol{\alpha} + \boldsymbol{\beta X_{ij}} + \boldsymbol{\mu_j} + \boldsymbol{\varepsilon_{ij}} \tag{2}$$

Where $\alpha$ is the fixed intercept, $\beta$ is the fixed effects, $X$ is the covariate, $\mu j$ is the random effect, $p_{ij}$ is the probability that an $i^{th}$ bovine shistosoma outcome within the $j^{th}$ cluster (origin), $\varepsilon ij$ is the error.

Efficacy of sedimentation technique to detect *S.bovis* egg with respect to postmortem examination was determined by calculating sensitivity, specificity, likelihood ratio (LR), positive, and negative predictive values using *epiR* package [48].

# Result

## Coprological examination

**Overall prevalence of S. bovis infestation.** Out of the total 768 faecal samples examined (384 from the field and 384 from the abattoirs) 16.7% (95% CI = 14.10–19.49) samples were found to be positive for *S.bovis* eggs and morphologically the eggs were oval to spindle shaped with centrally bulged and terminal spine (Fig 1).

**Risk factors analysis.** Variables with a p-value < 0.25 under the univariable mixed-effect logistic regression model such as breed, body condition and age of cattle were subjected to the multivariable mixed-effect logistic regression model (Table 1).

The final model for bovine shistosoma is presented in Table 2. It was assessed as to whether any interaction terms were needed to be incorporated into the model. This was examined by fitting each of the two-way interaction terms formed from all the explanatory variables, one at a time to the model that had all the main effects. Accordingly, there is no significant interaction effect (P >0.05) in the final model. Consequently, the final model included breed, body condition and age variables. Accordingly, local cattle had higher odds (OR = 2.44, 95% CI = 1.34–4.43) of being positive for shistosoma compared to crossbreed cattle. Poor body conditioned cattle had higher odds (OR = 4.09, 95% CI = 2.45–6.83) than good body condition cattle to be shistosoma positive. In addition, adults (OR = 1.78, 95% CI = 1.21–3.28) and old (OR = 1.39, 95% CI = 0.66–2.82) age cattle had higher odds of being positive for *S.bovis* than young age cattle.

The conditional $R^2$ value for the overall model was 0.16; indicating that 16% of the variance was explained by both the fixed and random effect. The marginal $R^2$ value for the fixed effects was 0.13, indicating that the fixed effects accounted for 13% of the variation in the data. The interclass correlation coefficient (ICC) for the random effect (origin) was 0.035, telling that 3.5% of the total variance in response outcome is accounted for by the clustering (Table 2).

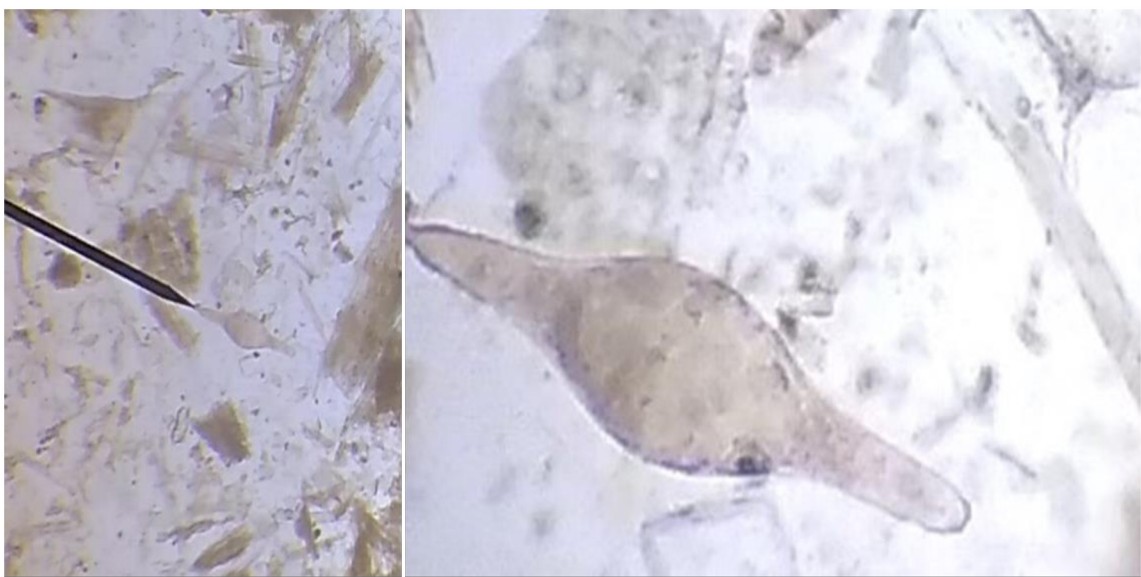

**Fig 1. Identified *Schistosoma bovis* egg using stereomicroscope.**

## Abattoir survey

**Diagnostic efficacy of sedimentation technique.** Out of 384 animals presented for slaughtering in the abattoir, eggs of the parasite were detected from 55 (14.3%) animals using coprological examination (sedimentation technique) and adult worms were recovered from 66 (17.2%) animals during necropsy. The evaluation of the relative efficiency of sedimentation technique with respect to postmortem examination for diagnosis of *S.bovis* was presented in Table 3. The sensitivity and specificity of sedimentation techniques, keeping postmortem examination as a reference test was 74.24% [61.99–84.22] and 98.11% [95.94–99.30], respectively. The post-test probability of S.bovis infestation (predictive value for the positive result of sedimentation technique) was 89.09% (95%CI, 78.50%– 94.81%), when compared to postmortem examination (Table 3).

**Table 1. Univariable mixed effect logistic regression risk factor analysis with prevalence of *S.bovis*.**

| Variable | Category | Total examined | No. of positive (%) | 95% CI (percentage) | OR (95% CI) | P-value |
|---|---|---|---|---|---|---|
| Sex | Male | 571 | 91 (15.90) | (13.16–19.16) | Ref. | |
| | Female | 197 | 37 (18.80) | (13.94–24.81) | 1.22 (0.80, 1.86) | 0.397 |
| Age | Young | 104 | 12 (11.53) | 6.72,19.09) | Ref. | |
| | Adult | 485 | 8 9 (18.35) | (15.16,22.03) | 1.72 (0.90,3.28) | 0.06 |
| | Old | 179 | 27 (15.08) | (10.58,21.06) | 1.36 (0.66,2.82) | 0.39 |
| Breed | cross | 169 | 14 (8.28) | (4.99,13.42) | Ref. | |
| | Local | 599 | 114 (19.03) | (16.09, 22.37) | 2.60 (1.45,4.67) | 0.001 |
| BCS | Good | 330 | 33 (10.00) | (7.21,13.71) | Ref. | |
| | Medium | 276 | 48 (17.39) | (13.37, 22.30) | 1.99 (1.69,4.01) | 0.004 |
| | Poor | 162 | 47 (29.01) | (22.57, 36.42) | 3.70 (2.25,6.08) | <0.001 |
| Total | | 768 | 128 (16.7) | (14.10–19.49) | | |

CI = Confidence Interval; OR = Odds Ratio; Ref. = Reference.

**Table 2. Multivariable mixed-effect logistic regression risk factor analysis with the prevalence of *S.bovis*.**

| Risk factor | Categories | No. of Animals | | OR (95% CI) | P-value |
|---|---|---|---|---|---|
| | | Examined | Positive (%) | | |
| Breed | Cross | 169 | 14 (8.3) | Ref. | |
| | Local | 599 | 114 (19.0) | 2.44 (1.34,4.43) | 0.003 |
| Body condition | Good | 330 | 33 (10.0) | Ref. | |
| | Medium | 276 | 48 (17.4) | 2.14 (1.33,3.46) | 0.002 |
| | Poor | 162 | 47 (29.0) | 4.09 (2.45,6.83) | < 0.001 |
| Age | Young | 179 | 12 (11.5) | Ref. | |
| | Adult | 485 | 8 9 (18.4) | 1.78 (1.21,3.28) | 0.02 |
| | Old | 104 | 27 (15.1) | 1.39 (0.66,2.82) | 0.24 |
| | Overall P. | 768 | 128 (16.7) | | |

CI = Confidence Interval; OR = Odds Ratio; Ref. = Reference.

R2GLMM(c) = 0.16, R2GLMM(m) = 0.13; Intraclass correlation coefficient = 0.035.

**Adult worm recovery and gross pathological findings.** Of the 384 slaughtered cattle 17.2% (95%CI = 13.55–21.34) were found to be positive for adult worm in their mesenteric veins during necropsy, and then the adult worms (the oral sucker, acetabulum (ventral sucker) and gynecophoral canal) were characterized using stereomicroscopy (Fig 2).

In the present study, schistosoma lesions were found in the liver and intestinal tracts. The pathological characteristics were described below:

Liver: Most of the pathological findings were tiny whitish foci, which were scattered under the capsule and on the cut surface of the liver. Fibrosis and thickening of the portal tracts were observed in heavily infected animals, the liver was hardened and extensively fibrosed, showing multiple, elevated greyish nodules which consisted of greatly dilated portal veins occupied by hemorrhagic thrombi (Fig 3A and 3B).

The intestinal tracts: particularly the small intestine, presented a thickened, slightly hyper-emic mucosa, with focal petechial hemorrhages and occasional ulcerations. In a few heavily infected cattle, the mucosa was intensely congested and coated with mucous exudate. In some

**Table 3. Evaluation of the efficacy of sedimentation technique for the diagnosis of bovine schistosomiasis (post-mortem examination as reference test).**

| Test properties | | Postmortem examination | | |
|---|---|---|---|---|
| | | +ve | -ve | Total |
| Coprological examination (sedimentation technique) | +ve | 49 | 6 | 55 |
| | -ve | 17 | 312 | 329 |
| | Total | 66 | 318 | 384 |
| Sensitivity (%) [95%CI] | 74.24% [61.99–84.22] | | | |
| Specificity (%) [95%CI] | 98.11% [95.94–99.30] | | | |
| PLR [95%CI] | 39.35 [17.59–88.03] | | | |
| NLR [95%CI] | 0.26 [0.17–0.40] | | | |
| Disease prevalence | 17.19 [13.55–21.34] | | | |
| PPV [95%CI] | 89.09% [78.50–94.81%] | | | |
| NPV [[95%CI] | 94.83% [91.15–96.51%] | | | |
| Accuracy | 94.01% [92.41–96.17%] | | | |

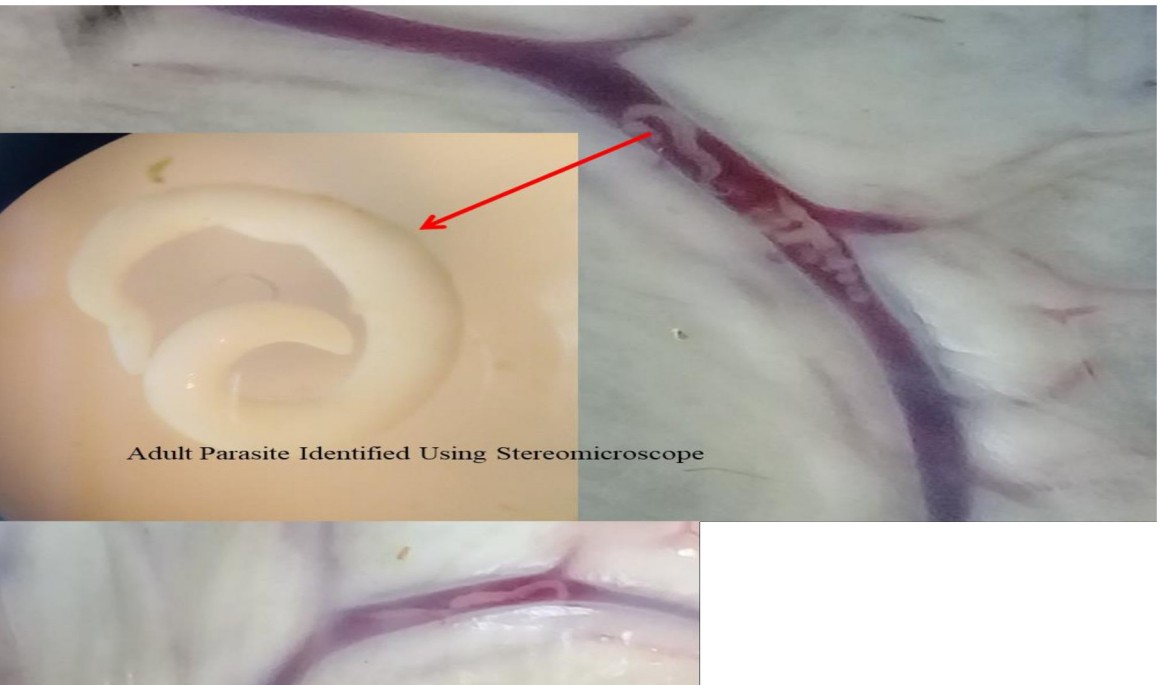

**Fig 2. Adult *S.bovis* in the mesenteric vein of cattle and it's morphological characteristics.**

animals superficial intestinal veins were occluded by thrombi, but more often, the vessels were engorged with blood, tortuous and thick-walled (Fig 3C and 3D).

## Discussion

This study quantified the magnitude of bovine shistosoma and its associated risk factors in cattle. Accordingly, the coprological examination both from the field and abattoirs showed that an overall prevalence of 16.7% (128/768), whereas a prevalence of 17.2% (66/384) was found based on postmortem examination of slaughtered cattle for schistosome adult worms in their mesenteric veins. The findings in the present study are in agreement with previous reports of Getachew et al. [49] in Kemissie and Yalelet [10] in Bahirdar who reported prevalence of 17.2% and 17.4% respectively. The results obtained both by coprological and postmortem examination are higher when compared with previous findings of Abebe et al. [11] 4.49% in Agaro; Almaz [29] 10.93% in Bahir Dar; Zelalem [50] 12.5%, Mengistu et al. [25] 10.17%, Mersha et al. [12] 13.70% in Fogera; and Mihret and Samuel [26] 7.6% in Debre Tabor. The probable reasons for the higher prevalence in the present study might be due to the presence of known water bodies (lakes) in the selected sites may favor the development and multiplication of intermediate hosts and practicing of free grazing on the pasture land might be another predisposing factor. In support of this, Urquhart et al. [51] has explained the importance of water bodies and marshy areas for the occurrence of schistosomiasis. This is also supported by the explanations of Jesus et al. [52] and Narcis et al. [53] that the prevalence and occurrence of schistosomiasis in a given area could be influenced by local climate conditions, presence of water reservoirs, lakes, rivers, and availability of suitable final and intermediate hosts.

In the other hand the present findings is lower than the previous studies conducted in Bahirdar by Amen et al. [24] 28%; in Bahirdar; Almaz and Solomon [54] 37.7% and in Dembia

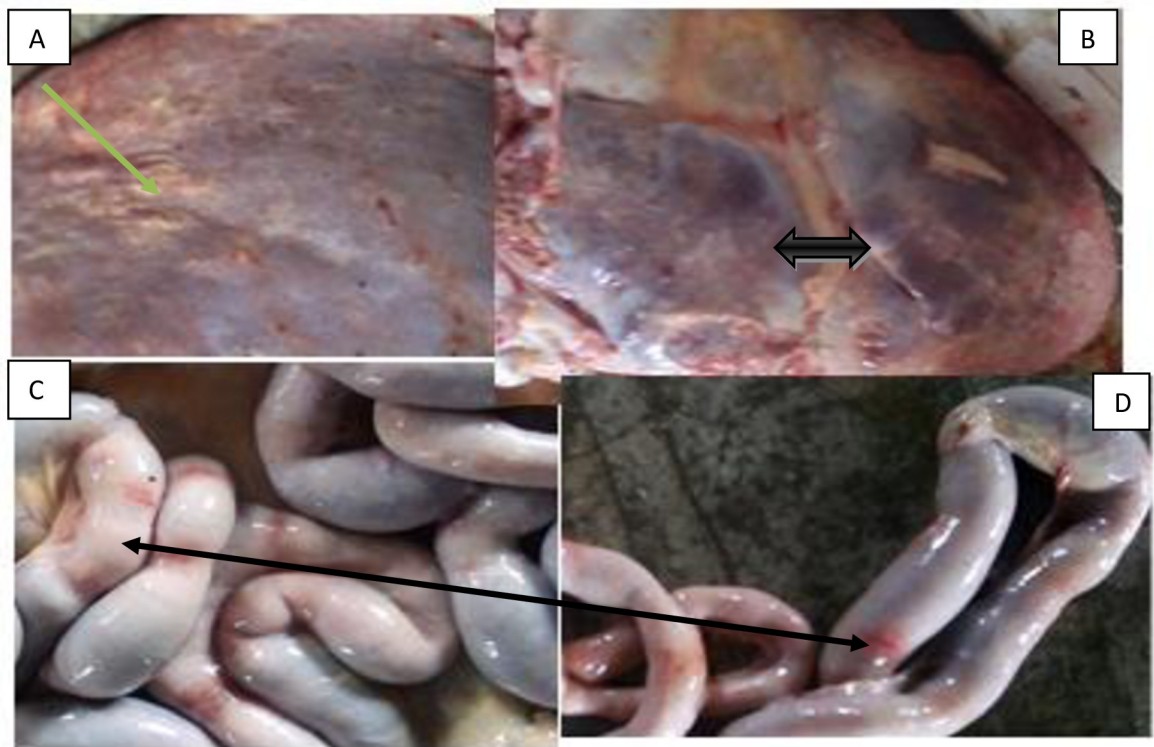

**Fig 3. Lesions induced by *S.bovis* in the liver and intestinal tract of cattle.** Lesions induced by *S.bovis* in the liver and intestinal tract of cattle. The single arrow in picture **"A"** presents hardened and extensively fibrosed multiple, elevated greyish nodules in the live and the short double arrow in picture "**B**' showed fibrosis and thickening of the portal tracts, the long double arrow in picture C and D indicates hemorrhagic enteritis.

by Alemseged [55] 27.13%. The reason for the lower prevalence might be in our study only single faecal sample examination is used in the diagnosis of parasites egg; this may be resulted in lower probability of detecting schistosoma eggs in faeces. This is also supported by Bushara et al. [56] trematodes are intermittent egg layers so that the chance of detecting eggs during single faecal sample examination may be minimal and difference in sampling seasons might be another factor.

In the present research work prevalence of *S. bovis* was significantly higher (P<0.005) in local breed animals than the cross breed animals. This finding is found to be in line with the previous prevalence studies of Belayneh and Tadesse [57] in Bahirdar Town; Samrawit [58] in Bahir Dar and its surrounding areas and Alemayehu and Mulat [59] in Dangila district. This might be due to the fact that local breed cattle were kept in outdoor and repeatedly exposed for Schistosoma infection than the crossbred animals, because crossbreds are kept for the purpose of milking in semi-intensive or intensive management systems with the provision of quality concentrate and roughages feed as well as clean water [60].

In the current study *S. bovis* infestation showed statistically significant variation (P<0.005) among the three body condition categories. Higher prevalence was recorded in poor body condition animals as compared to medium and good body condition animals. This finding in agreement with the previous results reported by Abebe et al. [11]; Lulie and Guadu [61]; Merawe et al. [62]; Belayneh and Tadesse [57]; Mihiret and Samuel [26] and Miressa and Feyissa [63] who confirmed that the infection increased with animals which had a poor body

condition score. This could be that the acquired immunity status of poor body condition and weak animals becomes more suppressed and susceptible, which might be due to malnutrition and other concurrent infections. Moreover, S. bovis infection can result in weight gain loss and weak acquired immunity [64].

The current findings showed that age was significantly related (P<0.005) to the occurrence of *S. bovis* infection in the studied animals. This is in agreement with reports of many researchers in Ethiopia and across the world [11, 25, 62, 65–67] who stated a significant effect of age on *S. bovis* infection in the studied animals and higher prevalence was recorded in adult and old aged animals. This could be attributed to the fact that adult and old cattle groups cover large areas and have high grazing capacity than young age groups under extensive and semi intensive management system, where the prevalence of cercariae infection is predominant. Besides, the adult worms of *S.bovis* were recovered from mesentery veins and the pathology of *S. bovis* was also studied during postmortem examination. Schistosome lesions were characterized in the liver and intestinal tracts. The current finding agrees with the reports of Hussein et al. [68] and Almaz et al. [69] where the recovered adult worms of *S.bovis* only limited on mesenteric veins compared to portal and urinary bladder veins. Generally, the prevalence of bovine schistosomiasis recorded in this study based on coprological and postmortem examination revealed that *S.bovis* is one of the endemic diseases contributing to loss in productivity and production of cattle in the study areas that need serious attention in the future.

## Conclusions

Bovine Schistosoma is prevalent in the study area which causes a significant impact on the production of cattle. For effective prevention and control of the disease, adult local cattle with medium and poor body conditions should be prioritized for deworming and future surveillance. The liver and intestinal tracts are the main organs location where Schistosoma pathological lesions are found. Therefore; well-planned deworming activities should be practiced to reduce the exposure of animals to Schistosoma and the impact of schistosome infection in the study districts and direct killing of intermediate host snails with chemicals or destroying their habitats through drainage systems should be implemented. Additionally, awareness should be created for the owners of the animals about the risk factors of the disease and the mode of transmission.

## Supporting information

**S1 Dataset.**
(SAV)

**S2 Dataset.**
(SAV)

**S1 File. R_script.**
(R)

**S2 File. Operational definition of body condition.**
(DOC)

## Acknowledgments

The authors are grateful to Wollo University and Kombolcha regional laboratory administrator for allowing us to conduct the laboratory work and for providing information. We also

appreciate and thank district and abattoir administrators of the study sites for their unreserved support during the data collection process.

## Author Contributions

**Conceptualization:** Gashaw Molla, Tarekegn Tintagu, Bethelehem Alemu.

**Data curation:** Gashaw Molla, Tarekegn Tintagu, Kassahun Tadesse.

**Formal analysis:** Tarekegn Tintagu, Alula Alemayehu Assen.

**Investigation:** Gashaw Molla, Tarekegn Tintagu.

**Methodology:** Gashaw Molla, Tarekegn Tintagu.

**Project administration:** Tarekegn Tintagu.

**Resources:** Tarekegn Tintagu.

**Supervision:** Tarekegn Tintagu, Ahmed Yasin.

**Validation:** Tarekegn Tintagu.

**Visualization:** Kassahun Tadesse.

**Writing – original draft:** Gashaw Molla, Ahmed Yasin, Bethelehem Alemu.

**Writing – review & editing:** Gashaw Molla, Tarekegn Tintagu, Alula Alemayehu Assen.

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
