## [Decision Letter · Decision Letter 0]

23 Nov 2021

PONE-D-21-33189STUDY ON BOVINE SCHISTOSOMIASIS IN SOME SELECTED AREAS OF SOUTH WOLLO AND OROMIA ZONES OF AMHARA REGION, NORTH-EAST ETHIOPIAPLOS ONE

Dear Dr. Tarekegn Tintagu Gizaw,

Thank you for submitting your manuscript to PLOS ONE. After careful consideration, we feel that it has merit but does not fully meet PLOS ONE’s publication criteria as it currently stands. Therefore, we invite you to submit a revised version of the manuscript that addresses the points raised during the review process.

We look forward to receiving your revised manuscript.

Kind regards,

A. K. M. Anisur Rahman, Ph.D.

Academic Editor

PLOS ONE

Journal Requirements:

2. Thank you for submitting the above manuscript to PLOS ONE. During our internal evaluation of the manuscript, we found significant text overlap between your submission and the following previously published works, some of which you are an author.

https://www.iiste.org/Journals/index.php/JBAH/article/view/35397

https://www.tandfonline.com/doi/abs/10.1080/00034983.1975.11687004

https://www.journalofscience.org/index.php/GJSFR/article/download/1992/1853/

https://tci-thaijo.org/index.php/tjvm/article/download/73778/59528

Please revise the manuscript to rephrase the duplicated text, cite your sources, and provide details as to how the current manuscript advances on previous work. Please note that further consideration is dependent on the submission of a manuscript that addresses these concerns about the overlap in text with published work.

● A clean copy of the edited manuscript (uploaded as the new *manuscript* file

"There was no finacial funder to this research work"

"All the authors declared that there is no competing interest "

7. PLOS requires an ORCID iD for the corresponding author in Editorial Manager on papers submitted after December 6th, 2016. Please ensure that you have an ORCID iD and that it is validated in Editorial Manager. To do this, go to ‘Update my Information’ (in the upper left-hand corner of the main menu), and click on the Fetch/Validate link next to the ORCID field. This will take you to the ORCID site and allow you to create a new iD or authenticate a pre-existing iD in Editorial Manager. Please see the following video for instructions on linking an ORCID iD to your Editorial Manager account: https://www.youtube.com/watch?v=_xcclfuvtxQ

Additional Editor Comments:

In addition the comments of the two reviewers please address the following while revising the manuscript:

1. Please merge study design and study population under materials and methods.

2. Data analysis: Use of Chi-square test only is not enough for the identification of risk factors. The animals are clustered within herd and districts. In this type of data, the mixed-effects logistic regression analysis should be preferred over chi-square test. Initially, the authors should run univariable mixed effect logistic regression model considering district as random intercept [univariable screening]. Then variables with a p-value <0.25 in the univariable screening should be included in the multivariable mixed-effect logistic regression analysis. The authors should describe the methods of model selection, model fit, collinearity, confounding and interaction.

3. The authors tested 384 samples by using both coprological and postmortem methods. So, the authors can easily evaluate the performance [sensitivity, specificity, positive predictive value and negative predictive value] of coprological method considering postmortem as a gold standard. The will add some new results which have diagnostic importance.

Reviewers' comments:

Reviewer's Responses to Questions

**Comments to the Author**

1. Is the manuscript technically sound, and do the data support the conclusions?

Reviewer #1: Yes

Reviewer #2: Partly

2. Has the statistical analysis been performed appropriately and rigorously? 

Reviewer #1: Yes

Reviewer #2: Yes

3. Have the authors made all data underlying the findings in their manuscript fully available?

Reviewer #1: Yes

Reviewer #2: Yes

4. Is the manuscript presented in an intelligible fashion and written in standard English?

Reviewer #1: Yes

Reviewer #2: No

5. Review Comments to the Author

Reviewer #1: In this study, the authors highlight Schistosoma bovis in cattle relating to its the prevalence and to characterize pathological lesions induced by Schistosoma bovis including risk factor assessment for the animal-level infestation of South Wollo and Oromia Administrative Zones, Amhara regional state, Ethiopia via a cross sectional survey. The study shows the occurrence of S. bovis infection with the overall prevalence of 16.7 % and 17.2 % using fecal sample examination and postmortem examination, respectively. The infection rate was statistically significantly associated with the body condition score, breeds and origins of the studied animals, but not with the sex and age of sampled animal. The overall outlook of the study is interesting. However, there are some lacunae in this manuscript that are needed to address before publication of the manuscript. Details included in the attached file (comments).

Reviewer #2: There are few published articles from the sites included in this study. However, this provides a through characterization of S. bovis from multiple sites. Although the paper contains useful information, I have several concerns that should be addressed before the paper is ready for publication.

General:

Please carefully check the scientific names and make them italic.

I strongly recommend to improve the laboratory coprological examination section with brief description of the laboratory technique.

Please provide necessary references and explanations against the Assumptions made to describe the findings.

Abstract:

1. “In addition to the coprological examination postmortem examination was carried out at Kombolcha and Dessie municipal abattoirs on 384 cattle which came from various livestock markets from the selected districts to recover the adult worms from mesenteric veins and to characterize the gross lesions in liver and intestinal tracts”. Sentence is too long and difficult to understand and realize the meaning. Please rewrite the sentence in break it down in two three sentences.

2. Of the total of 768 faecal samples examined 128 (16.7 %) were found positive for Schistosoma bovis eggs. Provide the 95% CI for the positive percentage.

3. Findings of the abattoir survey showed that 55(14.3%) cattle were positive for schistosoma parasite egg and 66 (17.2%) cattle were found to be positive for adult worm in their mesenteric veins, these findings showed that there was strong measure of agreement (Kappa=0.775) between coprological and post-mortem examination. Provide the 95% CI for the positive percentage.

Introduction:

1. Ethiopia has the largest number of livestock in Africa, [1-2]. Livestock contribute to over 40% of the value of annual agricultural production and no less 15% the gross national product (GNP). Provide few information including total cattle population in the study sites.

2. However, the livestock sector is not efficiently and fully exploited due to several constraints like malnutrition, traditional husbandry practice, poor genetic makeup and prevailing diseases [3-4]. Add few more information about availability or limitations of the veterinary services to diagnose and treat the animal diseases in Ethiopia.

3. In the introduction, write a paragraph on different clinical signs found at different stage of Schistosomiasis in cattle.

4. Therefore, considering the economic importance of the disease, agro ecological variation, different animal husbandry practices and disease prevention and control strategy of the five districts, the current study was designed to determine the prevalence of bovine schistosomiasis (S. bovis), to identify potential risk factors for the occurrence bovine schistosomiasis and to characterize pathological lesions induced by Schistosoma bovis infection. The study objectives is not specific. Please rewrite the study objectives including the study sites name.

Materials and Methods:

1. The study was conducted in some selected districts of South Wollo Zone (Kalu, Tehuledere and Ambasel Districts) around Ardibo, Logo and Golbo lakes and Borkena river) and Oromia Zone at Cheffa valley. Those study areas were selected purposively based the availabilities of permanent water bodies. The word ‘some” is vague. Specify the no. of districts included in the study.

2. Fig 1: From the study map, Amhara district is looking abnormally bigger than others. Please check it and make it clearer to the readers.

3. The study animals were cattle in the field and also the cattle that presented for slaughtering in Dessie and Kombolcha municipal abattoirs from purposely selected study areas without discrimination of their sex, age, origin, breed and body condition. It is quite difficult to understand the district name of Dessie and Kombolcha municipal abattoirs. Rewrite the sentence in which they are situated.

4. Study Design: A cross sectional study was conducted from November 2020 to June 2021 to determine the prevalence of bovine schistosomiasis and associated risk factors. “determine the prevalence of bovine schistosomiasis and associated risk factors” is part of objective. So remove this portion and rewrite the sentence.

5. The cattle population in each study district was obtained from local government veterinary records and the proportion of cattle from each district were weighted according to the estimated population size in that particular district. Create a table and enlist the district names with cattle population from corresponding district.

6. Both in the field and abattoirs the study animals were restrained properly and approximately 10 grams of fresh faecal sample was collected directly from the rectum using gloved hands and then the collected fecal samples were transported with cold chain to Wollo University, School of Veterinary Medicine, Parasitology laboratory. Write this sentences under a separate subheading “sample collection”.

7. What kinds of coprological methods author’s used for this paper is not clear. Please include the methods like direct microscopy or floatation or sedimentation etc. that you have used.

8. The test agreement between Coprological and post-mortem examination was analysed using Kappa statistics. Please mention the different kinds of agreements like weak, moderate, good and strong agreement with references.

9. An official letter of cooperation was obtained from Wollo University ethical clearance committee. Include the issuing no. of the official letter.

Results:

1. Provide 95% CI for all the prevalence or percentage values of S. bovis mentioned throughout the manuscript.

2. Association of Sex, Age, Breed and Body condition of the Study Animals with Schistosoma bovis infection

a. Revise the title as “Association of potential risk factors and Schistosoma bovis infection in cattles of south wollo and oromia zones of amhara region, north-east Ethiopia

b. For this section, firstly write the results with statistical significant values (p<0.005) and then write insignificant findings.

3. Figure 4: Use indicator to identify the specific lesions in the figures as you did for figure 3.

Discussion:

1. The probable reasons for the higher prevalence in the present study might be due to the presence of known water bodies (lakes) in the selected sites may favor the development and multiplication of intermediate hosts and practicing of free grazing on the pasture land might be another predisposing factor. As the authors has no grazing related information of their studied cattles , hence, cite a paper where there is evidence in favor of authors arguments on grazing influencing the prevalence of S. bovis.

2. This is also supported by the explanations of Jesus et al. [40] and Narcis et al. [41] that the prevalence and occurrence of schistosomiasis in a given area could be influenced by local climate conditions, presence of water reservoirs, lakes, rivers, and availability of suitable final and intermediate hosts. In earlier paragraph, you argued that the higher prevalence might be due to the presence of known water bodies (lakes) in the selected sites may favor the development and multiplication of intermediate hosts and now you are giving explanation against that which is not possible. Please read few more papers related to your findings and rewrite the explanation with references.

3. The association of potential host and environmental factors with the occurrence of S.bovis in the selected study sites were assessed and identified. I did not find any environmental factors that you have explored through this study. Please check it.

4. In the present research work prevalence of schistosoma bovis was statistically significantly higher (P<0.005) in local breed animals than the cross breed animals. Write schistosoma bovis as S. bovis and make it italic.

5. This might be due to the fact that local breed cattle were kept in outdoor and repeatedly exposed for Schistosoma infection than the cross breed animals, because cross breeds are kept for the purpose of milking in semi-intensive or intensive management systems with the provision of quality concentrate and roughages feed as well as clean water. Please add references supporting your opinion.

Conclusion and recommendations:

1. Without enlisting the recommendations write them in a single paragraph.

6. PLOS authors have the option to publish the peer review history of their article (what does this mean?). If published, this will include your full peer review and any attached files.

Reviewer #1: **Yes: **Sk Shaheenur Islam, Department of Livestock Services, Bangladesh.

Reviewer #2: No

---

## [Author Response · Author response to Decision Letter 0]

10 Mar 2022

Dear Respected academic editors and reviewers 

Thank you for giving us the opportunity to submit a revised draft of the manuscript “Bovine Schistosomiasis in Some Selected Areas of South Wollo and Oromia Zones of Amhara Region, North-East Ethiopia” We appreciate the time and effort that you providing feedback on our manuscript and we are also grateful for the insightful comments and valuable improvements to our paper. Considering both the academic editors and reviewers comments all section of the manuscript are thoroughly revised and mistakes are corrected. Those changes are highlighted with yellow color and track changed in the revised manuscript.

Response for the academic editor 

Dear Respected academic Editor 

1. We thank you so much for giving us the chance to revise our manuscript. As you explained that our first submitted manuscript was not fully meet PLOS ONE’s publication criteria, we apologies for the inconvenience, but now we have tried to follow the format. 

2. Regarding the text overlap of our submission and the previously published works, we have admitted that but now we have revised and cited our sources as we can as possible. 

3. As per your comment the study design and study population under materials and methods has merged (line no 168-176)

4. Data analysis: we really thank you so much for your high level professional comments and advise. As per your comments by considering of the district as random intercept the data is analysed using univariable mixed effect logistic regression and then variables with a p-value <0.25 in the univariable screening included in the multivariable mixed effect logistic regression analysis. We have also described the methods of model selection, model fit, collinearity, confounding and interaction. Data analysis about the abattoir survey using coprological and postmortem methods, we have also evaluated the efficacy/performance of sedimentation technique by calculating it’s sensitivity, specificity, positive predictive value and negative predictive value] considering postmortem as a gold standard test. 

Response for reviewer #1

Title: The comment give on the title is accepted by the authors and the first title “Study on Bovine Schistosomiasis in Some Selected Areas of South Wollo and Oromia Zones of Amhara Region, North-East Ethiopia” rephrases to “Bovine Schistosomiasis in Some Selected Areas of South Wollo and Oromia Zones of Amhara Region, North-East Ethiopia”, the word “study on” is removed.

Overall structure of the manuscript: The comments forwarded by the respected reviewer about the structure of the manuscript are fully accepted and now corrected as per the guidelines of the journal. Throughout the manuscript scientific names are thoroughly checked and italized. 

Abstract: All the correction measures given by the reviewer are accepted and the authors tried to shorten and modified the whole section of the abstract. 

Introduction: 

As suggested by the reviewer, we have added and reviewed the livestock statistics/data in the first paragraph of the introduction section (Line no 51-60) 

The reviewer kindly requested us to add few more lines in this paragraph “Parasitic diseases have been indicated among the major constraints in animal production throughout the tropical and sub-tropical countries of the world. Trematodes in general and schistosoma in particular have been reported as one of the major problems both in animals and humans around the world [5-7]”. As per the comments of the reviewer we have modified and merged with the paragraph highlighted in green color. The geographical distribution of bovine schistosomiasis has been determined primarily by the distribution of snail intermediate host particularly Bulinus species which are important for the occurrence of disease in bovine species. It has been indicated as endemic in sub-Saharan Africa and also found commonly in northern, eastern, southwestern and central parts of Ethiopia [8-10]. The endemicity of the disease particularly in the area with large permanent water bodies and marsh pasture area which anticipated for being ideal for the propagation and maintenance of the intermediate host (snails) has been found responsible for high prevalence of the infestation [11-12]. Out of 10 species reported to naturally infested cattle, six have received particular attention mainly because of their recognized veterinary significance; Schistosoma bovis, S.indicum, S. japonicum, S. matthei, S. intercalatum, S. nasale and S. rodhoni . S. bovis, S. matthei and S. intercalatum are the most important species that can cause schistosomiasis in ruminants [7]. (line no 76-87)

As suggested by the reviewer the phrase “schistosoma infection” is changed into “schistosoma infestation” not only in the introduction part but also throughout the manuscript

As per the recommendation of the reviewer the paragraph highlighted in red color “The prevalence of Schistosoma bovis has been reported from different regions of the country by a number of authors; 28% in Kemissie by Ameni et al. [17] in Fogera it was 10.17% by Mengistu et al. [18], and 13.70% by Mersha et al. [16], in Debre Tabor it was 7.6% by Mihret and Samuel 19], in Dangila it was 11.5% by Alemayehu and Mulat [20] and in and around Bahir Dar it was 26.3% by Samuel et al. [21] were evident by coproscopic examination. The report from Bahir Dar Abattoir by Hailu [22] revealed that prevalence 48%; 30.3% by Yalelet [15] and 28.14 by Almaz [23] 28.14% and in Jimma 13.46% prevalence by Abebe et al. [24]’ Paraphrased into the paragraph highlighted by green color “The prevalence of Schistosoma bovis has been reported from different regions of Ethiopia. Accordingly, prevalence of 28% in kemisse [24], 10.17% in Fogera [25], 13.70% in Fogera [12], 7.6% in Debretabor [26], 11.5% in Dangila [20], 26.3 % in Bahir Dar [27], 48% in Bahir Dar [28], 30.3% in Bahir Dar [10], 28.14% in Bahir Dar [29] and 13.46% in Jimma [30] have been reported both by coprological and postmortem examinations” (line no 118-123 of the manuscript). 

Materials and methods 

 Study area description: as per the recommendation of the reviewer 1 to avoid vague words like “some selected districts”; authors have tried to correct the sentence “The study was conducted in the selected districts of South Wollo Zone such as Kalu, Tehuledere and Ambasel Districts around Ardibo, Logo and Golbo lakes and Borkena river) and Oromia special Zone of Dawa Cheffa and Artuma furci”

Study population: the reviewer 1 forwarded comment to put the operational definition of poor, medium and good body condition animal classification as a supplementary file, therefore, we have fully agreed with the reviewer comment and prepared as supplementary file (S2)

Sample size determination and sampling techniques: as per the suggestion of the reviewer to make clear these sentences under this sub- section, we have revised and rewrite again (line no 179-189)

Abattoir survey: as per the comment of the reviewer 1 to put reference for the gross lesion characterization, we have characterized the gross lesion using a method (visualization, palpation and incision) as described by Di Provvido et al. (2018) (line no 219 reference 41)

Ethical considerations: the reviewer kindly requested us to incorporate the number and date for ethical approval taken, therefore; we have been received the ethical clearance from Wollo University ethical clearance committee a letter dated on 12, October, 2020 with a reference number of WU/CMHS13/105/2020 (line no 226) 

Data management and analysis: for the measurement of agreement between sedimentation technique and post-mortem examination the reviewer advised us to use Cohen’s kappa statistics, therefore; the authors have accepted the recommendation and analysed using it and the efficacies of the two tests are also calculated. In addition to this as per the recommendation of the academic editors the data is analysed both by univariate and multivariate mixed effect logistic regression analyses. (In the revised version of the manuscript whole section of “Data management and analysis” is modified (line no 233-278)). 

Result 

As per the recommendation of the editors and reviewers major modification has made on the result section. 

Prevalence of S. bovis infestation: the reviewer 1 recommended us to demonstrate the prevalence rate (percentage) with 95% CI, as per the recommendation of the reviewer we have calculated 95% CI for all percentages and presented in table 1. 

As suggested by the reviewer 1 the title “Association of sex, age, breed and body condition of the Study Animals with Schistosoma bovis infection” is modified and shortens into “Risk factor analysis” (line no 288) 

Abattoir survey: 

In the comparison between sedimentation technique and postmortem examination as the value of kappa described by McHugh, (2012), the reviewer recommended us to change the strong measure of agreement into substantial agreement and we have corrected it and P-value is calculated (described from line no 338-345 and table 4). 

With the same to the academic editor the reviewer 1 recommended us to calculate sensitivity (Se) and specificity (Sp) as well as positive and negative predictive values (PPV and NPV) of two tests (sedimentation technique) and postmortem examination), therefore; for better presentation of study result we have calculated the efficacy of sedimentation technique (Se, Sp, PPV and NPV) by considering postmortem examination method as a gold standard (described from line no 348- 354 and table 5). 

Conclusions: 

As per the recommendation of the reviewer we have paraphrased the conclusion section. 

Author’s contributions: we have followed the journal style. 

Response for reviewer #2

General comments: as per the comments of the reviewer 2 we have thoroughly checked the whole document and italized the scientific names

Abstract: 

As the reviewer 2 commented us the description regarding the abattoir survey is too long and difficult to understand and realize the meaning; therefore; by accepting the comments given by the reviewer 2 we the authors have tried to revised and make clear to the readers (line no 23-25) 

The reviewer also commented us to provide 95% CI for the overall prevalence and the measure of agreement between coprological and post-mortem examination. The comment forwarded by the reviewer welcomed by the authors and the 95% CI is calculated and included in the manuscript. 

Generally we have made major modification on the abstract section by considering the comments forwarded by the editors and reviewers. 

Introduction: 

In paragraph one of this sections the same to reviewer 1 the reviewer 2 also recommended us to incorporate the total cattle population. Therefore; we the authors full agreed with the thoughts of the reviewer and included Ethiopian’s livestock population and reviewed again (line no 51-60). 

Supported with references we the authors have been explained that livestock sector in Ethiopia is not efficiently and fully exploited due to several constraints like malnutrition, traditional husbandry practice, poor genetic makeup and prevailing diseases. But the reviewer 2 commented us to add few more information about the availability or limitations of the veterinary services to diagnose and treats the animal diseases in Ethiopia. As per the recommendation of the reviewer 2 we have reviewed and included information about availability or limitations of the veterinary services to diagnose and treat the animal diseases (line no 64-71). 

In addition the reviewer 2 also commented us to write a paragraph on different clinical signs found at different stage of schistosomiasis in cattle and the comment is well accepted and the clinical signs in the three stages of schistosomiasis reviewed and incorporated in the revised manuscript (line no 88-101)

Objectives:

 The reviewer 2 commented us the study objectives are not specific and needs modification. Comment is accepted and tried to modify and make them specific including the name of the study sites (line no 128-132)

Materials and Methods:

Study area description: The same to the first reviewer the second reviewer also recommend to avoid vague words like “some selected districts”, therefore; authors have tried to correct the sentence as “The study was conducted in the selected districts of South Wollo Zone such as Kalu, Tehuledere and Ambasel Districts around Ardibo, Logo and Golbo lakes and Borkena river) and Oromia special Zone of Dawa Cheffa and Artuma furci” 

Study map: The reviewer 2 commented us on the map (Fig 1) as the Amhara region is looking abnormally bigger than others. Considering the comment forwarded by the reviewer we have checked and remapped, attached in fig1. 

Study population: The reviewer is forwarded his comment as it is quite difficult to understand the district name of Dessie and Kombolcha municipal abattoirs, where they are situated? We the authors are fully agreed with the reviewer comment and both Dessie and Kombolcha municipal abattoirs are described under the abattoir survey sub-heading (line no 211-213).

Study Design: Reviewer 2 suggested us objectives should not be included in the study design, therefore; authors accepted the comment given by the reviewer and objectives already removed from this section. But as per the recommendation of the editor study design and population are merged as a one sub-heading (line no 169-176)

Data collection and processing

As per the recommendation of the reviewer 2 “Data collection and processing” sub-heading is modified into two separate sub-headings which are sample collection (line no 196) and parasite egg detection (line no 201) 

Coprological method: The reviewer 2 also recommended us to specify the kinds of coprological methods used for this paper. We sincerely appreciate the valuable comments and suggestions given by the reviewer. We have used the sedimentation technique and explained under sub-heading of “parasite egg detection” (line no 202-208)

Ethical considerations:

In the draft manuscript we have indicated that an official letter of cooperation was obtained from Wollo University ethical clearance committee but under this sub-heading the reviewer 2 suggested us to include the issuing no. of the official letter. Therefore, as the suggestion of the reviewer we have included the issue no of the official letter which is WU/CMHS13/105/2020 (line no 226). 

Data analysis: As the test agreement between coprological and post-mortem examination was analysed using Kappa statistics but the reviewer recommended us to mention the different kinds of agreements like weak, moderate, good and strong agreement with references. As per the reviewer recommendation we mentioned that the values ≤ 0 as indicating no agreement and 0.01–0.20 as none to slight, 0.21–0.40 as fair, 0.41– 0.60 as moderate, 0.61–0.80 as substantial, and 0.81–1.00 as almost perfect agreement and with a P value 0.05 was considered as statistically significant (McHugh, 2012) (line no 274-278). In addition to this the risk factor identification was analysed using univariate and multivariate mixed effect logistic regression (line no 235-270) and efficacy of sedimentation method (Se, Sp, PPV, NPV) were calculated considering postmortem examination as a golden standard (line no 271-274)

Result

As per the recommendation of the editors and reviewers authors have made major modification in the result section. 

Prevalence or percentage values of S. bovis: The reviewer suggested us to provide 95% CI for all the prevalence or percentage values of S. bovis mentioned throughout the manuscript. We think this is an excellent suggestion. We have calculated 95% CI for all percentage values of S.bovis and presented in table 1. 

Association of potential risk factors and Schistosoma bovis infestation: as per the suggestion of both the reviewer 1 and 2 this sub-heading is modified and shorten into “risk factor analysis” (line no 288) and risk factors which have statistical significant values (p<0.005) are explained first and followed by the insignificant values (line no 289-295)

Figure 4: The reviewer recommended us to use indicator to identify the specific lesions in the figures as we did for figure 3. Thank you for pointing this out, we have indicated the specific lesions both in the liver and intestinal tracts (figure 4) 

Discussion: 

In our draft manuscript we have wrongly given the same explanation/justification for both the lower and higher prevalence records when compared to the previously reported findings, which is commented by the reviewer 2. We thank you for insightful comment. Considering the comments forwarded by the reviewer we have carefully seen and provided logical and scientific explanation how the present prevalence records is higher and/or lower in comparison with previously reported findings and cited appropriately, which is marked and track changed in the revised manuscript (line no 400-404 for higher prevalence and 407-412 for lower prevalence record)

In the draft manuscript we have indicated that as the association of potential host and environmental factors with the occurrence of S.bovis were assessed and identified. But the reviewer 2 complained that as he did not find any environmental factors that we have explored through this study. However; we have been considered that the origin of the animals as an environmental factor, but we have convinced by the thought of the reviewer and description about the environmental factors is removed from the revised manuscript. 

The reviewer recommended us to put references supporting our opinion regarding the justification that we have given how local breed is stastically significant and crossbreed is not. As per the recommendation of the reviewer we have put supportive reference in the revised manuscript (line no 420 reference number 61)

Conclusion and recommendations: 

The same to reviewer 1 the reviewer 2 also suggested us to put in a single paragraph without enlisting the recommendations. We are fully agreed with both reviewer and corrected accordingly.

---

## [Decision Letter · Decision Letter 1]

25 Apr 2022

PONE-D-21-33189R1Bovine Schistosomiasis in some Selected Areas of South Wollo and Oromia zones of Amhara Region, North-East EthiopiaPLOS ONE

Dear Dr. Gizaw,

Thank you for submitting your manuscript to PLOS ONE. After careful consideration, we feel that it has merit but does not fully meet PLOS ONE’s publication criteria as it currently stands. Therefore, we invite you to submit a revised version of the manuscript that addresses the points raised during the review process.

 Academic Editor's comments:Lines 23-27:Please rephrase as suggested below:An abattoir survey was carried out on 384 cattle to evaluate the performance of the sedimentation method. The risk factors were identified using multivariable mixed-effect logistic regression analyses.Please report here the overall prevalence of coprological examination results which also includes the abattoir part. Then report the true prevalence of schistosomiasis based on postmortem results.Please delete  lines 31-41: starting from  "The prevalence" up to "body condition animals"Please rewrite this part by interpreting the odds ratio of three significant variables [breed, body condition, and age] in the multivariable model.Please conclude based on the risk factors. Like, local cattle with medium and poor body condition should be prioritized for deworming and future surveillance.Please delete lines 233-235 up to the citation of R: As univariable mixed-effect logistic regression analyses were performed to assess the association between the outcome and explanatory variables, the Chi-square test is not needed.Line 237 and elsewhere in the manuscript: Please replace univariate with univariable and multivariate with multivariable.Please cut lines 243-250 and paste them at the beginning of the data management and analysis.Please delete lines 273-274: Also, the likelihood ratio (LR) was determined for evaluating the validity of the sedimentation technique using the epiR package.Please add the likelihood ratio in the previous sentence and cite 48 as all of these parameters were estimated using a function of this package. Please delete lines 274 [starting from the "The test] to 278. It is no longer needed when the performance of the sedimentation technique was evaluated which is more informative just than just testing the agreement. Please delete lines 289-295. Please add here the results of the univariable screening. Just mention the name of variables included in the multivariable model citing Table 1. Please merge Tables 1 and 2. It can be done by deleting the last two columns of Table 1 and replacing them with the last two columns of Table 2.The odds ratio estimate shown in Table 2 for sex and age are incorrect.For example, the prevalence of schistosomiasis in male cattle is 15.9%. The male category was considered as a reference [1]. However, the prevalence of schistosomiasis in female cattle is 18.8% higher than that of male cattle. So, the odds ratio for female cattle must be >1 but the authors presented <1. This is also the same for age. In addition, it is better to consider the "young" category of age as a reference for having the lowest prevalence.Please delete this sentence from lines 320-321. They have a significant effect (p<0.05) on bovine Schistosoma.Please delete this sentence from lines 324-325:  Furthermore, young cattle had lower odds (OR=0.64, 95% CI, 0.30-1.35) than old cattle to be shistosoma positive. Because the estimation of the odds ratio for the age group was not correct and also the mentioned odds ratio was not significant. Please recalculate and consider young age as a reference.Please rename the title of Table 5: Evaluation of the efficacy of sedimentation technique for the diagnosis of bovine schistosomiasis (postmortem examination as reference test)Please delete this sentence and Table 4: The findings obtained from the current work showed that there was substantial agreement (Kappa = 0.775) between coprological (sedimentation technique) and post-mortem examination.Please delete lines 431-435 as gender was not significantly associated.Lines 436-445: Here authors made a self-contradictory discussion. Table 3 shows age as a significantly associated variable [although the mentioned odds ratio is not correct] but they are discussing it as non-significant.Please rephrase the long sentence in lines 446-448.Please rephrase the sentence in lines 449-452 [starting from "The current"]Please delete the lines 453-459 as these are results. Please discuss here the six false-positive coprological examination results.lines 467-475:Please rewrite the conclusion based on the results as also suggested earlier.

Kind regards,

A. K. M. Anisur Rahman, Ph.D.

Academic Editor

PLOS ONE

Journal Requirements:

Reviewers' comments:

Reviewer's Responses to Questions

**Comments to the Author**

1. If the authors have adequately addressed your comments raised in a previous round of review and you feel that this manuscript is now acceptable for publication, you may indicate that here to bypass the “Comments to the Author” section, enter your conflict of interest statement in the “Confidential to Editor” section, and submit your "Accept" recommendation.

Reviewer #1: All comments have been addressed

Reviewer #2: All comments have been addressed

2. Is the manuscript technically sound, and do the data support the conclusions?

Reviewer #1: Yes

Reviewer #2: Yes

3. Has the statistical analysis been performed appropriately and rigorously? 

Reviewer #1: Yes

Reviewer #2: Yes

4. Have the authors made all data underlying the findings in their manuscript fully available?

Reviewer #1: Yes

Reviewer #2: Yes

5. Is the manuscript presented in an intelligible fashion and written in standard English?

Reviewer #1: Yes

Reviewer #2: Yes

6. Review Comments to the Author

Reviewer #1: The manuscript substantially improved. However, typo-editing errors (scientific name, odd ratio, p value where appropriate) present throughout the text and tables. Author(s) should fix this issue before publication. Additionally, authors should check the plagiarism issue in the final version.

Reviewer #2: Thanks to the authors, they have addressed all the inquiries. Now the manuscript is looking better. I have no further inquiries.

7. PLOS authors have the option to publish the peer review history of their article (what does this mean?). If published, this will include your full peer review and any attached files.

Reviewer #1: No

Reviewer #2: No

---

## [Author Response · Author response to Decision Letter 1]

22 May 2022

Academic Editor's comments:

Lines 23-27:

Please rephrase as suggested below:

An abattoir survey was carried out on 384 cattle to evaluate the performance of the sedimentation method. The risk factors were identified using multivariable mixed-effect logistic regression analyses.

Please report here the overall prevalence of coprological examination results which also includes the abattoir part. Then report the true prevalence of schistosomiasis based on postmortem results.

Thank you for your suggestions and comments forwarded. The suggested comment were included in the improved manuscript from L24-29.

Please delete lines 31-41: starting from "The prevalence" up to "body condition animals"

Please rewrite this part by interpreting the odds ratio of three significant variables [breed, body condition, and age] in the multivariable model.

Thank you for the suggestion and comment. We have deleted line 31-41 of the manuscript written from L31-34. 

Please conclude based on the risk factors. Like, local cattle with medium and poor body condition should be prioritized for deworming and future surveillance.

We are grateful for the suggestions and included the suggestion in the improved manuscript from L37-38.

Please delete lines 233-235 up to the citation of R: As univariable mixed-effect logistic regression analyses were performed to assess the association between the outcome and explanatory variables, the Chi-square test is not needed.

Thank you for the suggestions and we did the same as per the suggestions in the improved manuscript.

Line 237 and elsewhere in the manuscript: Please replace univariate with univariable and multivariate with multivariable.

We replaced univariate with univariable and multivariate with multivariable as per the suggestions in the improved manuscript.

Please cut lines 243-250 and paste them at the beginning of the data management and analysis.

Thank you for the suggestion. We cut and paste the stated statement in the improved manuscript from L214-221.

Please delete lines 273-274: Also, the likelihood ratio (LR) was determined for evaluating the validity of the sedimentation technique using the epiR package.

Please add the likelihood ratio in the previous sentence and cite 48 as all of these parameters were estimated using a function of this package. 

We have deleted the lines and included the suggestions in the improved manuscript from L248-250.

Please delete lines 274 [starting from the "The test] to 278. It is no longer needed when the performance of the sedimentation technique was evaluated which is more informative just than just testing the agreement. 

Thank you very much for the suggestion and we did the same in the improved manuscript.

Please delete lines 289-295. Please add here the results of the univariable screening. Just mention the name of variables included in the multivariable model citing Table 1. 

Thank you for the comments and we have included the suggestions in the improved manuscript from L260-265.

Please merge Tables 1 and 2. It can be done by deleting the last two columns of Table 1 and replacing them with the last two columns of Table 2.

We have merged the two tables and the merged table was presented in L268-270

The odds ratio estimate shown in Table 2 for sex and age are incorrect.

For example, the prevalence of schistosomiasis in male cattle is 15.9%. The male category was considered as a reference [1]. However, the prevalence of schistosomiasis in female cattle is 18.8% higher than that of male cattle. So, the odds ratio for female cattle must be >1 but the authors presented <1. This is also the same for age. In addition, it is better to consider the "young" category of age as a reference for having the lowest prevalence.

Thank you for the comments and suggestions. We have corrected the comment as per the suggestion in the improved manuscript version in L268-270.

Please delete this sentence from lines 320-321. They have a significant effect (p<0.05) on bovine Schistosoma.

We have deleted the lines as per the suggestions in the improved manuscript

Please delete this sentence from lines 324-325: Furthermore, young cattle had lower odds (OR=0.64, 95% CI, 0.30-1.35) than old cattle to be shistosoma positive. Because the estimation of the odds ratio for the age group was not correct and also the mentioned odds ratio was not significant. Please recalculate and consider young age as a reference.

Thank you for the suggestions and we have recalculated the odds ratio of the age category in the improved manuscript from L279-281

Please rename the title of Table 5: Evaluation of the efficacy of sedimentation technique for the diagnosis of bovine schistosomiasis (postmortem examination as reference test)

We have renamed the title of Table 3 in the improved manuscript

Please delete this sentence and Table 4: The findings obtained from the current work showed that there was substantial agreement (Kappa = 0.775) between coprological (sedimentation technique) and post-mortem examination.

We are grateful for the suggestions. We have deleted Table 4 and the lines as per the suggestion in the improved manuscript

Please delete lines 431-435 as gender was not significantly associated.

Thank you for the suggestion. We have deleted the lines in the improved manuscript

Lines 436-445: Here authors made a self-contradictory discussion. Table 3 shows age as a significantly associated variable [although the mentioned odds ratio is not correct] but they are discussing it as non-significant.

We are sorry for the discrepancies created and we have corrected the concern in the improved manuscript in L375-383

Please rephrase the long sentence in lines 446-448.

We have rephrased the sentence in the improved manuscript from L382-384

Please rephrase the sentence in lines 449-452 [starting from "The current"]

Thank you for the suggestion and we have corrected it as per the suggestions in L384-386.

Please delete the lines 453-459 as these are results. Please discuss here the six false-positive coprological examination results.

We have deleted the lines as per the suggestions and included the suggestions in L386-388

lines 467-475:

Please rewrite the conclusion based on the results as also suggested earlier.

The conclusion of the manuscript was written again considering the suggestions and comments earlier in the improved manuscript in L394-407

Review Comments to the Author

Reviewer #1: The manuscript substantially improved. However, typo-editing errors (scientific name, odd ratio, p value where appropriate) present throughout the text and tables. Author(s) should fix this issue before publication. Additionally, authors should check the plagiarism issue in the final version.

We the authors are indebted for the positive complement to the level of improvement of the manuscript. We have also made editions on the concerns raised. 

Reviewer #2: Thanks to the authors, they have addressed all the inquiries. Now the manuscript is looking better. I have no further inquiries.

We thank you very much

---

## [Editor Report · Decision Letter 2]

24 May 2022

PONE-D-21-33189R2Bovine Schistosomiasis in some Selected Areas of South Wollo and Oromia zones of Amhara Region, North-East EthiopiaPLOS ONE

Dear Mr. Gizaw,

Thank you for submitting your manuscript to PLOS ONE. After careful consideration, we feel that it has merit but does not fully meet PLOS ONE’s publication criteria as it currently stands. Therefore, we invite you to submit a revised version of the manuscript that addresses the points raised during the review process.

ACADEMIC EDITOR:Please rephrase Lines 35-38 as given below:Local cattle breed (OR= 2.44, 95%CI= 1.34-4.43), poor body condition (OR= 4.09, 95% CI=2.45-6.83) and adult (OR=1.78, 95% CI= 1.21-3.28) cattle are more likely to acquire schistosomiasis than crossbreed, good body condition, and young cattle.Similarly, please rephrase lines 41-42   In conclusion, adult local cattle with medium and poor body conditions should be prioritized for deworming and future surveillance.Lines 58-63: Please rephrase as, In addition, the existing veterinary service delivery under the current animal health system is considered to be unsatisfactory in both private and public sectors because of the low government attention, inadequate budget allocation, gaps in control of illegal drug circulation, implementation of ethical practices, and shortage of basic laboratory facilities [5].Please rephrase lines 77-81: Out of 10 species reported as naturally infesting cattle, six have received particular attention mainly because of their recognized veterinary significance: please mention those six with veterinary significance [7].Please delete lines: 267-270: "However, the sex of the cattle doesn’t have a significant association (P>0.05) with S. bovis. Accordingly, local breed cattle (OR=2.60; 95% CI= 1.45– 4.67) with poor body condition (OR= 3.70; 95% CI= 1.25–4.18) have a higher risk of acquiring the disease than crossbreed with good body condition cattle." Please start the discussion with the major findings and their importance to control bovine schistosomiasis.Line 363L Please delete "statistically"Please delete "rate" from lines 371, 372, and 376 as prevalence is a proportion [not a rate}.False positivity is related to specificity, not sensitivity. So, if no proper explanation exists then better to delete this sentence: "In this study, six (6) false positive coprological examination results were which might be to the sensitivity (74.24%) of the coprological test relative to postmortem examination."Please delete lines 400-401 as they are general statements, not conclusions: "This study aims to determine the prevalence and associated risk factors of S. bovis in cattle.  In addition, the pathological lesions caused by S. bovis were characterized. Accordingly,"Also, please delete these sentences as they are results, not conclusions: The disease is significantly associated with the age, breed, and body condition of the animals. As a result, adult local cattle with poor body conditions animals were at higher risk to acquire the disease.Please rephrase as, For effective prevention and control of the disease, adult local cattle with medium and poor body conditions should be prioritized for deworming and future surveillance.=====================

Kind regards,

A. K. M. Anisur Rahman, Ph.D.

Academic Editor

PLOS ONE
---

## [Author Response · Author response to Decision Letter 2]

30 May 2022

Academic Editor's comments:

Please rephrase Lines 35-38 as given below:

Local cattle breed (OR= 2.44, 95%CI= 1.34-4.43), poor body condition (OR= 4.09, 95% CI=2.45-6.83) and adult (OR=1.78, 95% CI= 1.21-3.28) cattle are more likely to acquire schistosomiasis than crossbreed, good body condition, and young cattle.

Thank you for your suggestions and comments forwarded. The suggested comment was included in the improved manuscript from L35-38.

Similarly, please rephrase lines 41-42: In conclusion, adult local cattle with medium and poor body conditions should be prioritized for deworming and future surveillance.

Thank you for the suggestion and incorporated it into the improved manuscript in L41

Lines 58-63: Please rephrase as, In addition, the existing veterinary service delivery under the current animal health system is considered to be unsatisfactory in both private and public sectors because of the low government attention, inadequate budget allocation, gaps in control of illegal drug circulation, implementation of ethical practices, and shortage of basic laboratory facilities [5].

Thank you for your suggestions and comments forwarded. The suggested comment was included in the improved manuscript from L58-61.

Please rephrase lines 77-81: Out of 10 species reported as naturally infesting cattle, six have received particular attention mainly because of their recognized veterinary significance: please mention those six with veterinary significance [7].

We are grateful for the suggestion and incorporated in the improved manuscript from L… to L76-78.

Please delete lines: 267-270: "However, the sex of the cattle doesn’t have a significant association (P>0.05) with S. bovis. Accordingly, local breed cattle (OR=2.60; 95% CI= 1.45– 4.67) with poor body condition (OR= 3.70; 95% CI= 1.25–4.18) have a higher risk of acquiring the disease than crossbreed with good body condition cattle."

Thank you for the comment and we have deleted the statement in the improved manuscript. 

Please start the discussion with the major findings and their importance to control bovine schistosomiasis.

Thank you for the suggestion and we started the discussion part with major findings in the improved manuscript in L331-335. 

Line 363L Please delete "statistically"

The mentioned word was deleted from the improved version of the manuscript.

Please delete "rate" from lines 371, 372, and 376 as prevalence is a proportion [not a rate}.

We have deleted the word from the improved manuscript version.

False positivity is related to specificity, not sensitivity. So, if no proper explanation exists then better to delete this sentence: "In this study, six (6) false positive coprological examination results were which might be to the sensitivity (74.24%) of the coprological test relative to postmortem examination."

We have deleted the statement from the improved manuscript version.

Please delete lines 400-401 as they are general statements, not conclusions: "This study aims to determine the prevalence and associated risk factors of S. bovis in cattle. In addition, the pathological lesions caused by S. bovis were characterized. Accordingly,"

We have deleted the lines as per the suggestion.

Also, please delete these sentences as they are results, not conclusions: The disease is significantly associated with the age, breed, and body condition of the animals. As a result, adult local cattle with poor body conditions animals were at higher risk to acquire the disease.

Thank you for the suggestion and we have deleted it from the improved manuscript version.

Please rephrase as, For effective prevention and control of the disease, adult local cattle with medium and poor body conditions should be prioritized for deworming and future surveillance.

Thank you for the suggestions and we have rephrased the statement in the improved manuscript version in L394-396.

---

## [Editor Report · Decision Letter 3]

1 Jun 2022

Bovine Schistosomiasis in some Selected Areas of South Wollo and Oromia zones of Amhara Region, North-East Ethiopia

PONE-D-21-33189R3

Dear Dr. Gizaw,

We’re pleased to inform you that your manuscript has been judged scientifically suitable for publication and will be formally accepted for publication once it meets all outstanding technical requirements.

Kind regards,

A. K. M. Anisur Rahman, Ph.D.

Academic Editor

PLOS ONE
---

## [Editor Report · Acceptance letter]

8 Jun 2022

PONE-D-21-33189R3 

Bovine Schistosomiasis in some Selected Areas of South Wollo and Oromia zones of Amhara Region, North-East Ethiopia 

Dear Dr. Gizaw:

I'm pleased to inform you that your manuscript has been deemed suitable for publication in PLOS ONE. Congratulations! Your manuscript is now with our production department. 

Kind regards, 

on behalf of

Dr. A. K. M. Anisur Rahman 

Academic Editor

PLOS ONE